# Probing Gut Participation in Parkinson’s Disease Pathology and Treatment via Stem Cell Therapy

**DOI:** 10.3390/ijms241310600

**Published:** 2023-06-25

**Authors:** Jea-Young Lee, Vanessa Castelli, Paul R. Sanberg, Cesar V. Borlongan

**Affiliations:** 1Center of Excellence for Aging and Brain Repair, Department of Neurosurgery and Brain Repair, University of South Florida Morsani College of Medicine, 12901 Bruce B Downs Blvd, Tampa, FL 33612, USA; jeayoung@usf.edu (J.-Y.L.); psanberg@usf.edu (P.R.S.); 2Department of Life, Health and Environmental Sciences, University of L’Aquila, 67100 L’Aquila, Italy; castelli.vane@gmail.com

**Keywords:** neurodegeneration, inflammation, dopaminergic depletion, cell transplantation, microbiome, gut-brain axis

## Abstract

Accumulating evidence suggests the critical role of the gut–brain axis (GBA) in Parkinson’s disease (PD) pathology and treatment. Recently, stem cell transplantation in transgenic PD mice further implicated the GBA’s contribution to the therapeutic effects of transplanted stem cells. In particular, intravenous transplantation of human umbilical-cord-blood-derived stem/progenitor cells and plasma reduced motor deficits, improved nigral dopaminergic neuronal survival, and dampened α-synuclein and inflammatory-relevant microbiota and cytokines in both the gut and brain of mouse and rat PD models. That the gut robustly responded to intravenously transplanted stem cells and prompted us to examine in the present study whether direct cell implantation into the gut of transgenic PD mice would enhance the therapeutic effects of stem cells. Contrary to our hypothesis, results revealed that intragut transplantation of stem cells exacerbated motor and gut motility deficits that corresponded with the aggravated expression of inflammatory microbiota, cytokines, and α-synuclein in both the gut and brain of transgenic PD mice. These results suggest that, while the GBA stands as a major source of inflammation in PD, targeting the gut directly for stem cell transplantation may not improve, but may even worsen, functional outcomes, likely due to the invasive approach exacerbating the already inflamed gut. The minimally invasive intravenous transplantation, which likely avoided worsening the inflammatory response of the gut, appears to be a more optimal cell delivery route to ameliorate PD symptoms.

## 1. Introduction

Parkinson’s disease (PD) manifests as a neurodegenerative disorder characterized by dopaminergic neuronal depletion in the substantia nigra pars compacta (SNpc) [1,2]. PD patients display motor symptoms, including tremor, rigidity, and bradykinesia, and non-motor symptoms, particularly gut motility [3,4]. Levodopa serves as the gold standard treatment for PD, but adverse side effects, such as dyskinesias, over the course of treatment complicate clinical outcomes [5].

Recognizing the limited efficacy of pharmacological treatments, which essentially offer palliative instead of disease-modifying outcomes, finding a novel therapy that retards or halts PD progression is an urgent clinical need [1]. Stem cell transplantation has emerged as a disease-modifying strategy for PD [6,7,8,9], acting via cell replacement and bystander effects, e.g., neurotrophic and anti-inflammatory factor secretion [10,11,12]. New evidence suggests that transplanted cells target the gut–brain axis (GBA) by dampening the gut’s inflammatory response and attenuating the neurodegenerative cell death cascades in the brain [13,14,15]. Indeed, our group has recently demonstrated that intravenous transplantation of human umbilical cord blood (hUCB) and plasma (P) reduced motor deficits, improved nigral dopaminergic neuronal survival, and dampened α-synuclein in multiple rodent and murine PD models, including 6-hydroxydopamine (6-OHDA), 1-methyl-4-phenyl-1,2,3,6-tetrahydro-pyridine (MPTP), and transgenic α-synuclein-overexpression [13,14,15]. To further test the direct involvement of the GBA in stem cell therapy, the present study directly implanted hUCB and P into the gut of transgenic α-synuclein-overexpressing mice (Figure 1). The results nullified our original hypothesis of enhanced therapeutic effects of intragut delivery of hUCB and P, in that transplanted transgenic α-synuclein-overexpressing mice displayed an exacerbation of motor and gut motility deficits and aggravated expression of inflammatory microbiota, cytokines, and α-synuclein in both their gut and brain. While the GBA significantly contributes to PD inflammation, directly targeting the gut for stem cell delivery worsens functional outcomes. Stem cell delivery via a minimally invasive approach, such as the intravenous route as shown in our previous studies [13,14,15], appears to be a safer and more effective cell delivery approach to ameliorate PD symptoms by circumventing the gut’s inherent inflammatory response to PD pathology.

## 2. Results

**Behavioral analyses.** To test whether intragut implantation enhances functional recovery, animals were tested in two motor tasks, namely rotarod and beam walk tests (Figure 2). Unexpectedly, results revealed that Wt + Tx and Tg + Tx significantly performed worse than their corresponding strains that received the vehicle, i.e., Wt + Vehicle and Tg + Vehicle (p’s < 0.05) in both tasks across post-transplantation testing days (rotarod: *p*  <  0.0001; F_9,32_  =  7.608; beam walk: *p*  <  0.0001; F_9,32_  =  7.396). Pairwise comparisons revealed that Tg mice performed worse than Wt mice (*p* < 0.05) across all time points in both tasks, with the transplanted groups clearly performing worse than their vehicle counterparts (p’s < 0.05), except on day 7 for the beam walk test, which showed that Wt + Tx exhibited significantly more severe motor balance than Tg + Vehicle (*p* < 0.05).

**Gut functional assays.** To assess whether intragut transplantation improved gut functions, we tested animals in colonic propulsion and gastric-emptying tasks. To our surprise, results revealed that Wt + Tx and Tg + Tx significantly performed worse than their corresponding strains that received the vehicle, i.e., Wt + Vehicle and Tg + Vehicle (p’s < 0.05) in both tasks (colonic propulsion: *p*  <  0.0001; F_3,8_  =  69.61; gastric emptying: *p*  <  0.0001; F_3,8_  =  403.8). Pairwise comparisons revealed that Tg mice performed worse than Wt mice (p’s < 0.05) in both tasks, with the transplanted groups obviously performing worse than their vehicle counterparts (p’s < 0.05).

**Gut microbiome analysis.** We used fluorescent in situ hybridization (FISH) analysis to reveal the inflammation-associated microbiota within the feces of the animals by focusing on probes for BAC303, EREC482, and LAB158, which we previously identified as sensitive to PD pathology [13,14,15] (Figure 2). In tandem with the motor and gut functional tests, our results surprisingly revealed that Wt + Tx and Tg + Tx significantly displayed highly elevated inflammation-associated microbiota than their corresponding strains that received the vehicle, i.e., Wt + Vehicle and Tg + Vehicle (p’s < 0.05) in all three microbiota (BAC303: *p*  <  0.0001; F_3,8_  =  106.7; EREC482: *p*  <  0.0001; F_3,8_  =  100.2; LAB158: *p*  <  0.0001; F_3,8_  =  72.61). Pairwise comparisons revealed that Tg + Vehicle, Tg + Tx, and Wt + Tx performed worse than Wt + Vehicle (p’s < 0.05) in all three microbiota examined here (p’s < 0.05).

**Histopathology of PD brain.** We next examined α-synuclein immunostaining in the brain (Figure 3), which revealed significantly elevated α-synuclein levels in Tg mice compared to Wt mice (*p* < 0.0001; F_3,8_  =  1937), with Tg + Tx significantly exhibiting the highest level of α-synuclein expression compared to the other treatment groups (p’s < 0.05). TH immunostaining (Figure 3) revealed that Tg mice showed a significantly more severe (*p*  <  0.0001; F_3,8_  =  694.4) dopaminergic depletion compared to Wt mice (p’s < 0.05). Moreover, inflammatory markers OX6 and TNF-α showed significantly increased recruitment of immune and inflammatory cells in the SNpc (Figure 3) (OX6: *p*  <  0.0001; F_3,8_  =  1592; TNF-α: *p*  <  0.0001; F_3,8_  =  1348) with the Tg + Tx mice significantly displaying the highest upregulation of inflammation compared to the other treatment groups (p’s < 0.05).

**Histopathology of gut mucosa.** We similarly assessed α-synuclein and inflammation levels in the gut (Figure 3). Results showed a significantly upregulated expression of α-synuclein and inflammatory markers in Tg mice compared to Wt mice (α-synuclein: *p*  <  0.0001; F_3,8_  =  218.5; OX6: *p*  <  0.0001; F_3,8_  =  591.6; TNF-α: *p*  <  0.0001; F_3,8_  =  2523), with Tg + Tx exhibiting the highest level of α-synuclein, OX6, and TNF-α expression compared to the other treatment groups (p’s < 0.05).

## 3. Discussion

Our findings revealed that intragut transplantation of stem cells worsened PD motor and non-motor symptoms coupled with upregulation of inflammation-relevant microbiota and cytokines in the gut and brain of transgenic mice. Additionally, such an invasive cell delivery approach even worsened the behavioral performance of wild-type animals, suggesting that, while the GBA is a critical source of inflammation in PD, direct implantation of stem cells into the gut may not improve, and may even exacerbate functional outcomes. One of the causes of the observed exacerbation can be ascribed to the fact that the direct transplantation of stem cells may alter the colonic barrier, allowing the passage of bacteria metabolites into the blood and worsening the motor symptoms [16]. Moreover, it has been reported that PD patients present a leaky gut and the transplantation of stem cells can even worsen the condition, thus impairing motor performances [16].

Pharmacological treatments such as Levodopa have shown robust clinical improvements in PD patients [5]. However, over time, the drug’s adverse effects limit the efficacy of these treatments [5]. In addition, pharmacological treatments only offer palliative rather than disease-modifying outcomes in PD [5]. Stem cell transplantation for PD has been shown safe and effective in laboratory studies, but with varying efficacy readouts in the clinic [17]. To this end, cell replacement and by-stander effects have been implicated as mechanisms underlying the therapeutic effects of stem cell therapy in PD and other neurological disorders [18,19]. Recent studies have suggested that transplanted stem cells may solicit the GBA signaling pathway by sequestering inflammatory signals originating from the gut, thereby dampening the inflammation-mediated neurodegenerative cascade of cell death processes in the brain [20,21,22].

We recently demonstrated that intravenous transplantation of stem cells reduced gut-inflammation-associated microbiota and cytokines, coinciding with the rescue of behavioral and histological deficits in PD animal models [13,14,15]. In an effort to probe a cause-and-effect interaction between the transplanted stem cells and the GBA, we have now examined the functional effects of direct implantation of stem cells into the gut. Unfortunately, the results nullified our hypothesis, with intragut-transplanted mice exhibiting worsened PD behavioral and histological symptoms and aggravated inflammatory microbiota and cytokines in the gut and brain.

Because the gut is already inflamed during the course of PD pathology, a traumatic intervention, such as the present transplantation into the superior mesenteric artery, likely compounded the gut’s inflammatory response to the disease progression. Indeed, the three microbiota that we previously identified as closely associated with gut inflammation, namely BAC303, EREC482, and LAB158, were all significantly elevated in transplanted Tg mice. Intragut transplantation of stem cells even elevated these three microbiotas in Wt mice, indicating the severe traumatic consequence of such an invasive cell delivery regimen. While hUCB cells reached the gut (data not shown), this intragut transplantation exacerbated deleterious gut microbiota, upregulating inflammatory cytokines, including OX6 and TNF-α, ultimately leading to aberrant accumulation of α-synuclein in both gut and brain. That gut inflammation precedes α-synuclein aggregation, leading to neurodegeneration, has been shown as a vicious cell death event in PD [23,24,25].

In summary, the present observations further implicate the participation of the GBA in PD pathology, with the gut-inflammation-associated microbiota contributing to α-synuclein over-expression and subsequent neurodegeneration, and eventually, to the disease’s behavioral and histological hallmarks. Directly implanting stem cells into the gut may not be beneficial in arresting the peripheral inflammatory response. Instead, a minimally invasive route of cell administration, such as the intravenous approach, seems safer and more effective in diminishing the GBA’s pathological impact on the disease process.

## 4. Methods

**Human umbilical cord blood cells and plasma.** A detailed description of hUCB and P preparation has been previously reported [15]. Saneron CCEL Therapeutics, Inc. (Tampa, FL, USA) processed, cryopreserved, and supplied hUCB and plasma samples that were utilized in this study.

**Animal preparation and transplantation.** The present study followed experimental procedures approved by the University of South Florida Institutional Animal Care and Use Committee (IACUC) and in adherence to the ARRIVE 2.0 guidelines [26]. Additionally, animal handling and surgical processes were done to minimize discomfort and pain, based on the ethical regulations of the European Communities Council (Directive 2010/63/EU, protocol #542/2019-PR). Blinding of investigators to the treatment conditions was carried out until study completion. We enrolled 14-month-old male C57BL/6NJ (wild type or Wt) and C57BL/6N-Tg (Thy1-SNCA)15Mjff/J mice (transgenic or Tg) (The Jackson Laboratory, Bar Harbor, ME, USA). The choice of the age of Tg mice was based on previous studies [15] that showed that such age corresponds to overexpression of the wild-type human α-synuclein, with validated phenotypic, PD-like progressive nigrostriatal dopamine depletion and motor deficits. The mice enrolled in this study had free access to food and water, were housed under normal conditions (20 °C, 50% relative humidity, and a 12 h light/dark cycle), and were randomly assigned to a group by a staff member not involved in the study: Wt + Vehicle (*n*  =  4), Wt + hUCB + P transplants or Wt + Tx (*n*  =  4), Tg + Vehicle (*n*  =  4), and Tg + Tx (*n*  =  4). Transplantation involved delivery of 0.4 × 10^6^ hUCB cells in 50 μL of plasma (Figure 1). To deliver the cells into the small intestine, we followed the microsurgery procedure that targets the superior mesenteric artery, which supplies blood to the small and large intestine [16].

**Behavioral tests (Rotarod and Beam Walk).** Detailed descriptions of the rotarod test and beam walk test have been previously reported [15]. The rotarod test used a rotating drum (IITC Life Science, Woodland Hills, CA, USA) to assess the animal’s ability to balance over three trials on a rotating rod with speeds starting at 4 rpm and accelerating to 40 rpm in 300 s. The beam walk test assessed forelimb and hindlimb function using semi-quantitative scoring of grades 0 to 3. Animals underwent both tasks at baseline (prior to transplantation) and at days 1, 3, and 7 post-transplantations.

**Distal colonic propulsion.** As previously described [15], the distal colonic propulsion recorded the expulsion time of glass bead (0.5-mm-diameter), which was inserted about 2 cm into the rat’s anus, into the distal colon. This test was conducted on day 7.

**Gastric emptying.** Additionally, as previously reported [15], a solution of charcoal (10%) and acacia gum (2%) was given via oral gavage to the animal on the last survival day. Following euthanasia, the animal’s intestine was harvested, and the length (cm) that the charcoal solution traveled was measured. This test was conducted on day 7.

**Tissue collection.** Animals were deeply anesthetized, then perfused transcardially with phosphate-buffered saline (PBS) and 4% paraformaldehyde in PBS [15]. The intestines and brains of these animals were harvested and fixed. Coronal cryosections (40 µm) were processed from analyses.

**Microbiome analysis.** Following our previous protocol [15], fecal microbiota was analyzed and identified using FISH. The FISH images were collected and analyzed at 40× with the aid of an Olympus FV1000 laser scanning confocal microscope with Fluoview SV1000 imaging software.

**Immunofluorescence.** As described previously [15], immunostaining for tyrosine hydroxylase (TH) (1:100 TH, AB152; Millipore, Burlington, MA, USA), α-synuclein (1:250, NBP2-15365; NOVUS, Centennial, CO, USA), MHC II (OX-6; NB100-65541; Novus Biologicals, Centennial, CO, USA) and TNF-α (ab6671; Abcam, Cambridge, MA, USA) was captured throughout the entire SNpc or gut. AlexaFluor 488 and 594 secondary antibodies were used. Image analyses were conducted using an Olympus FV1000 laser scanning confocal microscope equipped with Fluoview SV1000 imaging software. The exclusion of primary antibodies substituted with 3% normal horse serum in PBS served as the controls, which showed a lack of immunoreactivity.

**Statistical analysis.** Data were statistically analyzed using one-way analysis of variance (ANOVA) and subsequent post hoc Bonferroni’s test with statistical significance set at *p*  <  0.05 (GraphPad version 5.01). Based on our previous PD animal modeling experience, we estimated a 15% variation within each group [13,14,15]. This variance was similar between groups when statistically compared.

## Figures and Tables

**Figure 1 ijms-24-10600-f001:**
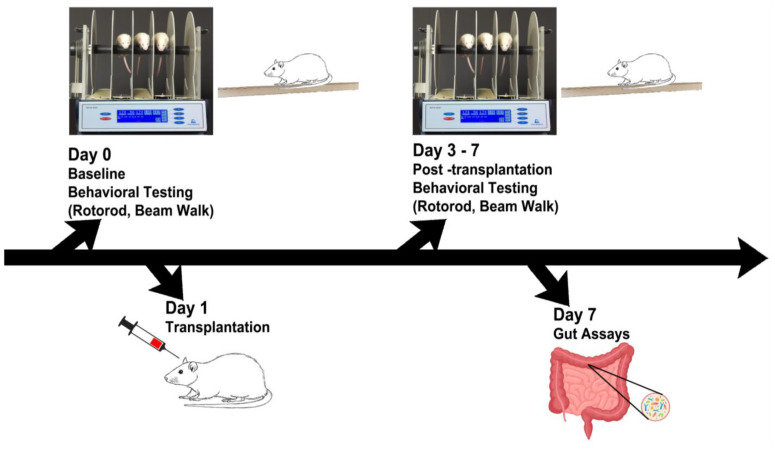
**Experimental design.** Behavioral tests were performed on days 0–7, followed by gut motility tests and microbiome assays, and immunohistochemical analyses of inflammatory cytokines and dopaminergic depletion on day 7.

**Figure 2 ijms-24-10600-f002:**
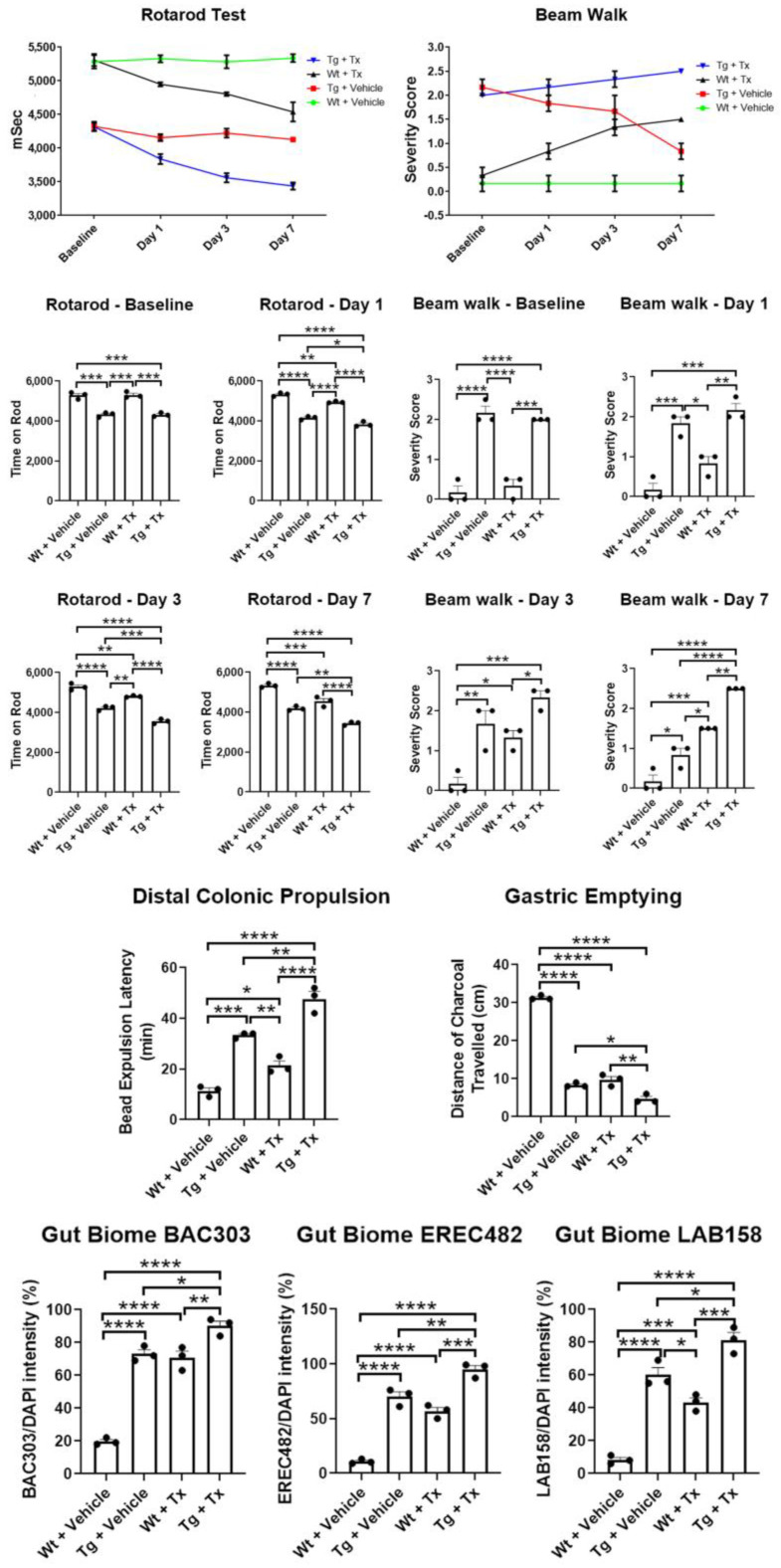
**Motor and non-motor deficits in experimental PD mice with stem cell therapy.** Direct implantation of stem cells to the gut exacerbated motor deficits in rotarod and beam walk tests with stem cell-transplanted mice performing significantly worse than vehicle-transplanted mice. Non-motor symptoms, specifically those associated with gut motility, were also worsened by intragut implantation of stem cells, as evidenced by significantly worsened outcomes in colonic propulsion and gastric emptying tasks by stem-cell-transplanted mice compared to vehicle-transplanted mice. The exacerbation of motor and non-motor symptoms in transplanted animals coincided with increased levels of gut-inflammation-relevant microbiota, BAC303, EREC482, and LAB158 (* p’s < 0.05; ** p’s < 0.01; *** p’s < 0.001; **** p’s < 0.0001).

**Figure 3 ijms-24-10600-f003:**
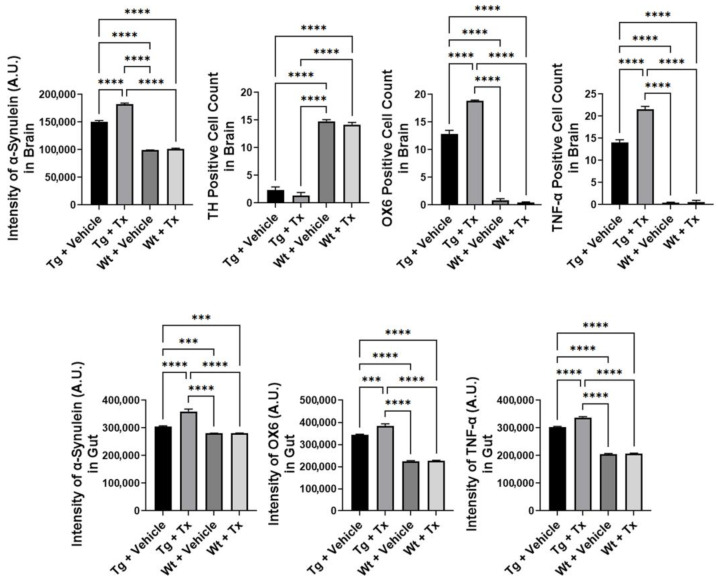
**Expression of α-synuclein and inflammatory markers in experimental PD mice with stem cell therapy.** Intragut stem cell transplantation significantly increased α-synuclein brain levels, with Tg + Tx significantly exhibiting the highest level of α-synuclein expression compared to the other treatment groups. Tg mice also displayed significantly more severe dopaminergic depletion compared to Wt mice. Additionally, inflammatory markers, OX6 and TNF-α, were significantly increased in the SNpc, with the Tg + Tx mice significantly displaying the highest upregulation of inflammation compared to the other treatment groups. Similarly, gut α-synuclein and inflammation levels were significantly upregulated in Tg mice compared to Wt mice, with Tg + Tx exhibiting the highest level of α-synuclein, OX6, and TNF-α expression compared to the other treatment groups (*** p’s < 0.001; **** p’s < 0.0001).

## Data Availability

All research data are available from Cesar V. Borlongan upon reasonable request.

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
