# Peer review of "Probing Gut Participation in Parkinson’s Disease Pathology and Treatment via Stem Cell Therapy"

_ijms, 2023, doi:10.3390/ijms241310600_

Round 1

Reviewer 1 Report

I read the manuscript entitled “Probing Gut Participation in Parkinson’s Disease Pathology 2 and Treatment Via Stem Cell Therapy” the authors studied the effect of intragut stem cell transplantation in a mice model, surprisingly resulting in worsening of motor performance and gut motility

The study is interesting and well performed; I have few comments.

In the discussion, it is possible that the direct transplantation of stem cells altered the colonic barrier allowing passage of bacteria metabolites into the blood worsening the motor symptoms. It is also possible that the resulting inflammation disrupted the blood-brain-barrier of mice allowing increased exposure to metabolites and inflammatory signals into the nervous system, increasing the motor symptoms, please comment on this and see:

 Baizabal-Carvallo JF, Alonso-Juarez M. The Link between Gut Dysbiosis and Neuroinflammation in Parkinson's Disease. Neuroscience. 2020;432:160-173. doi:10.1016/j.neuroscience.2020.02.030

 Minor points

As in this journal methods are at the end of the article, I suggest to enter the abbreviations early so we can read it in the order as it is, rather than first going to the methods section.

Author Response

Reviewer 1

  • I read the manuscript entitled “Probing Gut Participation in Parkinson’s Disease Pathology 2 and Treatment Via Stem Cell Therapy” the authors studied the effect of intragut stem cell transplantation in a mice model, surprisingly resulting in worsening of motor performance and gut motility. The study is interesting and well performed; I have few comments.

Response: We would like to thank Reviewer 1 for the positive comments and for the time spent in revising our manuscript. We appreciate the Reviewer’s comments and we tried to address all the points raised.

  • In the discussion, it is possible that the direct transplantation of stem cells altered the colonic barrier allowing passage of bacteria metabolites into the blood worsening the motor symptoms. It is also possible that the resulting inflammation disrupted the blood-brain-barrier of mice allowing increased exposure to metabolites and inflammatory signals into the nervous system, increasing the motor symptoms, please comment on this and see:

 Baizabal-Carvallo JF, Alonso-Juarez M. The Link between Gut Dysbiosis and Neuroinflammation in Parkinson's Disease. Neuroscience. 2020;432:160-173. doi:10.1016/j.neuroscience.2020.02.030

 Response: We appreciate the Reviewer comment, and we now added this point in the discussion section.

 Minor points

As in this journal methods are at the end of the article, I suggest to enter the abbreviations early so we can read it in the order as it is, rather than first going to the methods section.

Response: We thank the Reviewer for the suggestion.

Reviewer 2 Report

Review of a manuscript “Probing Gut Participation in Parkinson’s Disease Pathology and Treatment Via Stem Cell Therapy” by Jea-Young Lee and coauthors submitted to IJMS.

Parkinson's disease is a second after Alzheimer’s disease neurodegenerative disorder with a limited efficiency of disease-modifying pharmacological treatments.  There is a hope that emerging stem cell transplantation may be developed as a new approach for the cure of this disorder. This method may use cell replacement and neurotrophic and anti-inflammatory factor secretion. The authors investigated minimally invasive intravenous transplantation which might avoid worsening the inflammatory response of the gut and may represent an optimal cell delivery method. This is an important area of biomedical research, and the results presented in the manuscript will be interesting for the readers of the IJMS. 

The following corrections should be made.

Abstract

“in both gut and brain of rodent and murine PD models”. Classification of a mouse as a small rodent sometimes is debated and is controversial. It may be better to give a clearer definition of types of animals.

Introduction

Lines 36-37: After the sentence “Recognizing the limited efficacy of pharmacological treatments, which essentially offer palliative instead of disease-modifying outcomes, finding a novel therapy that retards or halts PD progression represents an urgent clinical need.” The authors should add a reference on a recent relevant review: ”Biomarkers in Parkinson’s Disease”. Chapter in a book Peplow PV et al.. (eds) Neurodegenerative Diseases Biomarkers. 2022. Neuromethods, v. 173. p 155-180. Humana, New York, NY. https://link.springer.com/protocol/10.1007/978-1-0716-1712-0_7

Results

Figure 1

The text under the animals in blue and red is too small and impossible to read.

Figure 2 It is unclear what the authors mean by saying “Intensity of alpha-synuclein (A.U) in gut? The meaning should be explained.

Line 87 “We used FISH analysis”. If the authors mean “Fluorescence in situ hybridization” they should give first the full name. 

Line 123. “Histopathology of gut mucosa. We similarly assessed α-synuclein and inflammation levels in the gut (Fig. 2).

Why the explanations to Figure 2 are placed after the Figure 3?

Discussion

Line 137: ”Motor and non-motor symptoms accompany PD [3, 4]” This sentence is not necessary in Discussion section, it is already used in the beginning of the Introduction.

 Methods

Lines 218-220. Immunofluorescence. As described previously [15], immunostaining for tyrosine hydroxylase (TH) (1:100 TH, AB152; Millipore, Burlington, MA), α-synuclein (1:250, NBP2 15365; NOVUS, Centennial, CO), MHC II (OX‐6; NB100‐65541; Novus Biologicals, Centennial, CO) and TNF‐α (ab6671; Abcam, Cambridge, MA)

Were secondary antibodies used for detection?

Author Response

Reviewer 2

Parkinson's disease is a second after Alzheimer’s disease neurodegenerative disorder with a limited efficiency of disease-modifying pharmacological treatments.  There is a hope that emerging stem cell transplantation may be developed as a new approach for the cure of this disorder. This method may use cell replacement and neurotrophic and anti-inflammatory factor secretion. The authors investigated minimally invasive intravenous transplantation which might avoid worsening the inflammatory response of the gut and may represent an optimal cell delivery method. This is an important area of biomedical research, and the results presented in the manuscript will be interesting for the readers of the IJMS. 

The following corrections should be made.

Response: We would like to thank Reviewer 2 for the positive comments and for the time spent in revising our manuscript. We appreciate the Reviewer’s comments and we tried to address all the points raised.

Abstract

“in both gut and brain of rodent and murine PD models”. Classification of a mouse as a small rodent sometimes is debated and is controversial. It may be better to give a clearer definition of types of animals.

Response: We thank the Reviewer for the comment and we totally agree. We now defined the type of animals.

Introduction

Lines 36-37: After the sentence “Recognizing the limited efficacy of pharmacological treatments, which essentially offer palliative instead of disease-modifying outcomes, finding a novel therapy that retards or halts PD progression represents an urgent clinical need.” The authors should add a reference on a recent relevant review: ”Biomarkers in Parkinson’s Disease”. Chapter in a book Peplow PV et al.. (eds) Neurodegenerative Diseases Biomarkers. 2022. Neuromethods, v. 173. p 155-180. Humana, New York, NY. https://link.springer.com/protocol/10.1007/978-1-0716-1712-0_7

Response: We appreciate the Reviewer’s comment and we added the reference as suggested.

Results

Figure 1

The text under the animals in blue and red is too small and impossible to read.

Figure 2 It is unclear what the authors mean by saying “Intensity of alpha-synuclein (A.U) in gut? The meaning should be explained.

Response: We appreciate the Reviewer’s comment and we modified the figure as suggested.

Line 87 “We used FISH analysis”. If the authors mean “Fluorescence in situ hybridization” they should give first the full name. 

Response: We appreciate the Reviewer’s comment and we added full meaning.

Line 123. “Histopathology of gut mucosa. We similarly assessed α-synuclein and inflammation levels in the gut (Fig. 2).

Why the explanations to Figure 2 are placed after the Figure 3?

Response: We apologize for the oversight, we now replaced the explanations.

Discussion

Line 137: ”Motor and non-motor symptoms accompany PD [3, 4]” This sentence is not necessary in Discussion section, it is already used in the beginning of the Introduction.

Response: We thank the Reviewer for the comment, we deleted the sentence as indicated.

 Methods

Lines 218-220. Immunofluorescence. As described previously [15], immunostaining for tyrosine hydroxylase (TH) (1:100 TH, AB152; Millipore, Burlington, MA), α-synuclein (1:250, NBP2 15365; NOVUS, Centennial, CO), MHC II (OX‐6; NB100‐65541; Novus Biologicals, Centennial, CO) and TNF‐α (ab6671; Abcam, Cambridge, MA)

Were secondary antibodies used for detection?

Response: We appreciate the Reviewer for the comment and we now provided all the necessary information.